# A New Dual-Mass MEMS Gyroscope Fault Diagnosis Platform

**DOI:** 10.3390/mi14061177

**Published:** 2023-05-31

**Authors:** Rang Cui, Tiancheng Ma, Wenjie Zhang, Min Zhang, Longkang Chang, Ziyuan Wang, Jingzehua Xu, Wei Wei, Huiliang Cao

**Affiliations:** 1Key Laboratory of Instrumentation Science & Dynamic Measurement, Ministry of Education, North University of China, Taiyuan 030051, China; cuirang@outlook.com; 2Tsinghua Shenzhen International Graduate School, Tsinghua University, Shenzhen 518055, China; mtc21@mails.tsinghua.edu.cn (T.M.); zhang-wj21@mails.tsinghua.edu.cn (W.Z.); 19955778426@163.com (J.X.); weiw20@mails.tsinghua.edu.cn (W.W.); 3School of Instrument Science and Opto-Electronics Engineering, Beijing Information Science & Technology University, Beijing 100192, China; 4College of Intelligence Science and Technology, National University of Defense Technology, Changsha 410073, China; changlongkang1999@126.com; 5Department of Electronic Engineering, Tsinghua University, Beijing 100084, China; wangziyu21@mails.tsinghua.edu.cn

**Keywords:** MEMS gyroscope, fault diagnosis platform, feature extraction

## Abstract

MEMS gyroscopes are one of the core components of inertial navigation systems. The maintenance of high reliability is critical for ensuring the stable operation of the gyroscope. Considering the production cost of gyroscopes and the inconvenience of obtaining a fault dataset, in this study, a self-feedback development framework is proposed, in which a dualmass MEMS gyroscope fault diagnosis platform is designed based on MATLAB/Simulink simulation, data feature extraction, and classification prediction algorithm and real data feedback verification. The platform integrates the dualmass MEMS gyroscope Simulink structure model and the measurement and control system, and reserves various algorithm interfaces for users to independently program, which can effectively identify and classify seven kinds of signals of the gyroscope: normal, bias, blocking, drift, multiplicity, cycle and internal fault. After feature extraction, six algorithms, ELM, SVM, KNN, NB, NN, and DTA, were respectively used for classification prediction. The ELM and SVM algorithms had the best effect, and the accuracy of the test set was up to 92.86%. Finally, the ELM algorithm is used to verify the actual drift fault dataset, and all of them are successfully identified.

## 1. Introduction

As an important device in navigation systems, gyroscopes can provide accurate navigation positioning and attitude parameters for carriers, and their accuracy and reliability play a rather important role in navigation systems, and are used in various aspects of spaceflight, aviation, and navigation [1]. With the development of modern technology, measurement and control systems are becoming more and more complex, resulting in difficulties regarding troubleshooting and increased possibility of failure. To improve the accuracy and reliability of the gyroscope, promptly providing navigation and positioning parameters for the carrier, detecting, identifying, and predicting navigation system faults, and guaranteeing the positioning accuracy and reliability of the navigation system are important tasks [2].

A fault diagnosis system needs to be able to perform three tasks: fault detection, to determine whether a fault has occurred in the system; fault isolation, to locate the fault and determine which sensor or actuator in the system has failed; and fault identification, to estimate the size, type or characteristics of the fault [3].

There are presently three primary categories of fault diagnosis technique: mathematical model-based diagnosis, input/output signal analysis and processing-based diagnosis, and artificial intelligence-based diagnosis.

Mathematical model-based fault diagnosis was first proposed by Dr. Beard of MIT in his thesis in 1971, and requires the establishment of a more accurate mathematical model of the research object in order to study the dynamic characteristics of the object and reproduce the control process at the theoretical level through mathematical means, thus achieving a reproduction and diagnosis of faults [4]. However, sometimes the mathematical model of the system is difficult to establish precisely, leading to a certain instability of the final diagnosis. The latter two types of fault diagnosis method do not require an accurate mathematical model to be provided, and also overcome redundant dependencies, leading them to be a hot topic at present for research in the field of gyroscope fault detection and diagnosis.

Signal processing-based methods are used to obtain monitoring signals by monitoring the process of the system, and to find the intrinsic relationship between the signals by extracting certain eigenvalues in the monitoring signals for fault detection [5]. Common methods include Fourier transform, wavelet transforms [6], empirical modal decomposition [7], and other methods. In [8], wavelet packet transform was used to extract the features of the satellite control moment gyroscope. Pu et al. [9] studied the application of Local Mean Decomposition (LMD) in depth for the diagnosis of mechanical faults in high-voltage circuit breakers.

Artificial intelligence-based fault diagnosis is an emerging research direction at present. It mainly relies on research results from current hot topics in the field of artificial intelligence, including support vector machines (SVM), fault trees, fuzzy logic, rough sets, neural networks, etc. Zhang et al. [10] proposed a new fusion fault diagnosis method based on FFT-based SAE and WPD-based NN for the fault diagnosis of gyroscopes; Zhao et al. [11] put forward a new CNN network scheme based on attention-enhanced convolutional blocks (AECB) for a data-driven CMG fault diagnosis scheme.

At present, it is quite difficult to perform diagnosis of complex faults due to the cost of gyroscope fabrication and the inconvenience of fault dataset acquisition, so in this study, a self-feedback development framework is proposed to design a novel dualmass MEMS gyroscope fault diagnosis platform. The platform has the following advantages:(1)The platform is a complete self-feedback system, integrating a dualmass MEMS gyroscope Simulink structure model and measurement and control system, gyroscope fault signal simulation model, data feature extraction and identification classification algorithm, and real data feedback verification.(2)The platform can generate seven types of signal: normal, bias, blocking, drift, multiplicity, period, and internal fault, and those signals are identified and classified using six algorithms—ELM, SVM, KNN, NB, NN, and DTA—after feature extraction, all of which have good diagnosis and recognition rates.(3)Although the neural network model can generally achieve good diagnosis and recognition rate when applied to fault diagnosis, there are also some problems, so the platform also reserves various algorithm interfaces for users’ independent programming, so that users can optimize the algorithm by themselves and connect the algorithm to this platform to verify the effectiveness of the algorithm.

## 2. Dualmass MEMS Gyroscope Fault Diagnosis Platform

### Overall Approach

Figure 1 depicts a novel dualmass MEMS gyroscope fault diagnosis platform that is based on the self-feedback development framework proposed in this paper. As illustrated in the figure, the platform consists of three key components: the gyroscope model, the theoretical model, and the platform model. The first component involves the analysis and modeling of the dualmass MEMS gyroscope’s structure, while the second component focuses on the analysis and modeling of the gyroscope’s measurement and control system [12,13,14]. The third component, the platform model, is primarily responsible for generating, extracting features from, identifying, and classifying various types of gyroscope fault. Subsequently, feedback verification of the experimental data is conducted, and this section provides a detailed explanation of the theoretical foundations of each of these three components.

## 3. The Principle of Dualmass MEMS Gyroscopes

### 3.1. Structural Model of Dual Mass Silicon Micromechanical Gyroscope

Dualmass MEMS gyroscopes are mainly composed of mechanically sensitive structures of silicon materials and driving and detection circuits [15,16]. Of these, the silicon structure possess a driving mode and a detection mode, and can convert the angular rate input signal into the displacement signal of the detection mode through the Coriolis effect. The driving circuit provides the necessary vibration conditions for the driving mode, and the detection circuit extracts the Coriolis signal. The gyroscope structure used is fully decoupled, as shown in Figure 2. In the absence of external impact and vibration, the kinetic equation of the gyroscope is described as:(1)$$\left[\begin{array}{l} m_{x}\;\;0\\ \;0\;\;m_{y} \end{array}\right]\left[\begin{array}{l} \ddot{x}\\ \ddot{y} \end{array}\right]+\left[\begin{array}{l} c_{xx}\;c_{xy}\\ c_{yx}\;c_{yy} \end{array}\right]\left[\begin{array}{l} \dot{x}\\ \dot{y} \end{array}\right]+\left[\begin{array}{l} k_{xx}\;k_{xy}\\ k_{yx}\;k_{yy} \end{array}\right]\left[\begin{array}{l} x\\ y \end{array}\right]=\left[\begin{array}{l} F_{dx}+2m_{p}\Omega_{z}\dot{y}\\ F_{dy}-2m_{p}\Omega_{z}\dot{x} \end{array}\right]$$

In the equation, *m_x_* and *m_y_* denote the equivalent quality of the drive mode and detection mode; *c_xx_*, *c_yy_*, *k_xx_*, and *k_yy_* are the effective damping and stiffness for the driving and detection modes, respectively; *c_xy_* and *k_xy_* are the damping and stiffness of the detection mode coupled to the driving mode; *c_yx_* and *k_yx_* are effective damping and stiffness for coupling the driving mode to the detection mode; *F_dx_* is the driving force; *F_dy_* is the feedback force of the detection mode; *M_p_* denotes the Coriolis mass; Ω*_z_* represents the angular rate around the z-axis; x and y represent the displacement in the driving and detection directions [17,18].

When the gyroscope structure is working, the driving frame will vibrate with constant frequency (resonant frequency of the driving mode) and amplitude along the x-direction. With an Ω*_z_* input, the Coriolis mass will drive the detection frame to move along the y-axis under the Coriolis force (which is related to Ω*_z_*). The mechanical thermal noise present in the gyroscope structure can be equivalent to a random, zero average Gaussian force, and its effect on the gyroscope can be equated to adding forces next to the dampers:(2)$$F_{n}=\sqrt{4K_{B}TCB}$$
where *F_n_* is the equivalent interference force of mechanical thermal noise; *K_B_* = 1.38 × 10^−23^ J/K is the Boltzmann constant; *T* is the absolute temperature; *C* and *B* is damping coefficient and working bandwidth of the gyroscope structure respectively. Due to the significant proportion of Coriolis mass in detecting equivalent mass, by substituting Equation (2) into Equation (1) with *m_y_ = m_p_*, we can obtain:(3)$$\left[\begin{array}{l} m_{x}\;\;0\\ \;0\;\;m_{y} \end{array}\right]\left[\begin{array}{l} \ddot{x}\\ \ddot{y} \end{array}\right]+\left[\begin{array}{l} c_{xx}\;c_{xy}\\ c_{yx}\;c_{yy} \end{array}\right]\left[\begin{array}{l} \dot{x}\\ \dot{y} \end{array}\right]+\left[\begin{array}{l} k_{xx}\;k_{xy}\\ k_{yx}\;k_{yy} \end{array}\right]\left[\begin{array}{l} x\\ y \end{array}\right]=\left[\begin{array}{l} F_{dx}+F_{nx}+2m_{p}\Omega_{z}\dot{y}\\ F_{dy}+F_{ny}-2m_{p}\Omega_{z}\dot{x} \end{array}\right]$$

From Equation (3), it can be analyzed that the driving mode is a second-order differential equation for forced vibration, and for the convenience of analysis, the driving displacement is set as:(4)$$x\left(t\right)=A_{x}\sin\left(\omega_{d}t\right)$$
where *A_x_* is the vibration amplitude of the driving mode; *ω_d_* is driving force frequency equal to *ω_x_*.

### 3.2. Dualmass MEMS Gyroscope Monitoring System

To realize a series of measures to improve gyroscope performance, such as quadrature correction and the detection of a closed loop, in this paper, modules for achieving these functions are added to the gyroscope model, and the simulation model diagram is displayed in Figure 3. The model contains the drive mode, the detection mode, a mutual coupling module between the drive and detection modes, a mechanical thermal noise module (this paper adopts the value of room temperature, 20 °C), a quadrature correction module, and a Ghosh homogeneous correction force module.

The model input signals are drive voltage *V_d_*, angular velocity input Ω*_z_*, detection feedback voltage *V_yb_*, and quadrature correction voltage *V_fs_*. The output signals include drive displacement signal *x*, detection displacement signal *y*, and detection modal force signal *F_q_*.

If the resonant frequencies $f_{x}=\frac{1}{2\pi }\omega_{x}=\frac{1}{2\pi }\sqrt{\frac{k_{xx}}{m_{x}}}$ and $f_{y}=\frac{1}{2\pi }\omega_{y}=\frac{1}{2\pi }\sqrt{\frac{k_{yy}}{m_{y}}}$ of the driving mode and the detection mode are set at 4050 Hz and 4060 Hz, respectively, in the model, the operating bandwidth *B* is approximately 5.4 Hz. Simultaneously, the quality factors $Q_{x}=\frac{\sqrt{m_{x}k_{xx}}}{c_{xx}}$ and $Q_{y}=\frac{\sqrt{m_{y}k_{yy}}}{c_{yy}}$ of both modes are set to 2000, and it is assumed that the equivalent input angular velocities caused by coupling stiffness and damping are 200 (°)/s and 5 (°)/s, respectively. The calculation method for equivalent input angular velocity can be obtained from the following two equations:(5)$$\left\{\begin{array}{l} -2\Omega_{k}m_{p}\dot{x}=xk_{yx}\\ -2\Omega_{c}m_{p}\dot{x}=xc_{yx} \end{array}\right.$$
where Ω*_k_* and Ω*_C_* are the equivalent input angular rates caused by coupling stiffness and coupling damping, respectively. Since the gyroscope structure often adopts a symmetrical structure, this article assumes that *k_yx_ = k_xy_*, *c_yx_ = c_xy_*.

The conversion of voltage and force in the model is achieved through comb capacitors and satisfies the equation:(6)$$\left\{\begin{array}{l} F_{static}=\frac{1}{2}\nabla C\left(x,y,z\right)V_{static}^{2}\\ C\left(x,y,z\right)=n\frac{\varepsilon hl}{d} \end{array}\right.$$
where *F_static_* is the electrostatic force; *C(x,y,z)* is the comb capacitance; *V_static_* is the voltage generated by electrostatic force; *n* is the number of comb teeth; *ε* is the dielectric constant; *h* is the structural thickness; *l* is the superimposed length of the comb capacitor; *d* is the spacing between comb capacitors [19,20].

### 3.3. Platform Model

A Simulink simulation model platform based on the gyroscope model and theoretical model is built, as shown in Figure 4. In this platform, the gyroscope measurement and control system and the gyroscope fault simulation model are designed, and then feature extraction and processing algorithms are designed by MATLAB/Simulink. The established model for the entire system was used for fault signal recognition and classification. The gyroscope fault simulation model was used to generate bias, blocking, drift, multiplicity, period, and internal fault signals, combined with normal signals for feature extraction, and classification was performed using six algorithms: ELM, SVM, KNN, NB, NN, and DTA. At the same time, the platform also reserves algorithm recognition interfaces for users to independently program various algorithms into the platform to prove the effectiveness of algorithms.

## 4. Algorithms

### 4.1. Variational Mode Decomposition (VMD)

The original signal *f*(*t*) was decomposed into *K* intrinsic mode components (IMF) via a constrained variational model, as shown in Equation (7):(7)$$\left\{\begin{array}{l} \underset{\left(u_{k})(w_{k}\right)}{\min}\left\{\sum_{k}{\left\|\partial_{t}[\sigma (t)+\frac{j}{t}u_{t}(t)]e^{-jw_{k}t}\right\|}_{2}^{2}\right\}\\ s.t\sum_{k}u_{k}=f(t) \end{array}\right.$$

In this context, {*u_k_*} = {*u*_1_, *u*_2_, …, *u_k_*} are the *k* IMFs, {*w_k_*} = {*w*_1_, *w*_2_, …, *w_k_*} denotes a series of center frequencies that were utilized in the decomposition process. Additionally, *σ_t_* represents the impulse function that was employed in the analysis.

The Lagrange operator *λ*(*t*) and the quadratic penalty factor α are introduced to transform the inequality constraint into an equation constraint, and the corresponding extended Lagrange expressions can be written as:(8)$$L(\left\{u_{k}\right\},\left\{w_{k}\right\},\lambda )=\alpha {\sum_{k=1}^{k}\left\|\partial_{t}\left[\left(\delta (t)+\frac{j}{\pi t}\right)\times u_{k}(t)\right]e^{iw_{k}t}\right\|}_{2}^{2}+{\left\|f(t)-\sum_{k=1}^{k}u_{k}(t)\right\|}_{2}^{2}+\langle \lambda (t),f(t)-\sum_{k=1}^{k}u_{k}(t)\rangle$$

The optimal solution to Equation (7) is obtained through the use of an alternate multiplier method to locate the saddle point of Equation (8). The iterative updating process for *u_k_* and *ω_k_* is shown in Equations (9) and (10), respectively. The analysis involved refining the intrinsic mode components (IMFs) and center frequencies *u_k_* and *ω_k_* through an iterative process. By utilizing the alternate multiplier method, the optimal solution to Equation (7) was obtained.
(9)$${\hat{u}}_{k}^{n+1}\left(\omega \right)=\frac{\hat{f}(\omega )-\sum_{i<k}{\overset{\wedge }{\underset{i}{u}}}^{n+1}(\omega )-\sum_{i<k}{\overset{\wedge }{\underset{i<k}{u}}}^{n+1}(\omega )+\frac{{\overset{\wedge }{\lambda }}^{n}(\omega )}{2}}{1+2\alpha {\left(\omega -\omega_{k}^{n}\right)}^{2}}$$
(10)$$\omega_{k}^{n+1}=\frac{\int_{0}^{\infty }\omega {\left|{\overset{\wedge }{\underset{k}{u}}}^{n+1}(\omega )\right|}^{2}d\omega }{\int_{0}^{\infty }{\left|{\overset{\wedge }{\underset{k}{u}}}^{n+1}(\omega )\right|}^{2}d\omega }$$

The value of *λ*(*ω*) was updated using the following equation:(11)$$\lambda^{n+1}(\omega )=\lambda^{n}(\omega )+\tau (f(\omega )-\sum_{k}^{n+1}u_{k}(w))$$

The above steps were repeated until the iteration stop condition was reached [21]:(12)$${\sum_{k}{\left\|u_{k}^{n+1}-u_{k}^{n}\right\|}_{2}^{2}/\left\|u_{k}^{n}\right\|}_{2}^{2}<\gamma$$

### 4.2. Sample Entropy (SE)

The size of sample entropy (SE) can be used to measure the self-similarity and complexity of the data sequence, and the calculation of SE does not depend on the data length of the signal, so it has better statistical stability and adaptability when quantifying the characteristics of the gyroscope fault signal [22]. The calculation steps of SE are as follows:

Step 1: Given the original time series *x* = {*x*_1_, *x*_2_, …, *x_N_*}, *m* is chosen as a suitable embedding dimension to construct a new state vector, which is *x* = {*x_i_*, *x_i+_*_1_, …, *x_i+m−_*_1_}, *i* = 1, *2*, …, *N − m*.

Step 2: The maximum difference distance between two state vectors is calculated as follows:(13)$$d(x_{i},x_{j})=\underset{k}{\max}(\left|x_{i+k}-x_{j+k}\right|)\;,k=0,1,\ldots ,m-1$$

Step 3: Given the similarity tolerance parameter *r*, *B_i_* is calculated, whose distance between *x_i_* and *x_j_* is less than or equal to *r*, and defined as follows:(14)$$B_{i}^{m}(r)=\frac{B_{i}}{N-m-1},1\leq i\leq N-m$$

Step 4: Calculate the mean and define *B^m^*(*r*):(15)$$B^{m}(r)=\frac{1}{N-m+1}\sum_{i=1}^{N-m+1}B_{i}^{m}(r)$$

Step 5: Increase the embedding dimension *m* + 1, and repeat the above four steps to calculate *B^m+^*^1^(*r*):(16)$$B^{m+1}(r)=\frac{1}{N-m}\sum_{i=1}^{N-m}B_{i}^{m+1}(r)$$

When the length of the time series is finite, *SE* can be calculated as follows:(17)$$SE(m,r,N)=-\ln\left(\frac{B^{m+1}(r)}{B^{m}(r)}\right)$$

### 4.3. Extreme Learning Machine (ELM)

The Extreme Learning Machine (ELM) represents a novel single-hidden-layer feedforward neural network architecture. One of its key features is the ability to set the parameters of hidden layer nodes either randomly or manually without the need for complex adjustments. Additionally, the learning process only involves a calculation of the output weight, significantly reducing computational complexity compared to other neural network models. By introducing a nonlinear activation function in the hidden layer, the ELM is capable of handling complex, nonlinear phenomena in an efficient and effective manner. Therefore, the convergence rate of the ELM algorithm is much faster than that of the traditional algorithm, because it does not require iteration. At the same time, random hidden nodes guarantee the global approximation ability [23]. Therefore, this paper proposes using ELM as a network model to predict the fault signal of the gyroscope. The network model of ELM is shown in Figure 5.

For the ELM network model training phase, N different input/output (*x_i_*, *t_i_*) are needed, which the *x_i_* = [*x_i_*_1_, *x_i_*_2_, …, *x_in_*]*^T^* ∈ *R^n^*, *t_i_* = [*t_i_*_1_, *t_i_*_2_, …, *t_im_*]*^T^* ∈ *R^m^.* The ELM network model, which comprises *L* hidden layer nodes, can be represented as follows:
(18)$$\sum_{i=1}^{L}\beta_{i}h\left(\omega_{i}\cdot x_{j}+b_{i}\right)=O_{j},\left(j=1\cdots N\right)$$
where *h*(*x*) is the hidden layer activation function; in ELM, the activation function provides nonlinear mapping for the system; *ω_i_* = [*ω_i_*_1_, *ω_i_*_2_, …, *ω_in_*]*^T^* is the input weight matrix; *β_i_* is the output weight matrix; *b_i_* is the *i*-th hidden layer bias; *ω_i_·x_j_* is the inner product of *ω_i_* and *x_j_*; *O_j_* represents the output of the model.

The goal of ELM network model training is to reduce the discrepancy between the target and the ELM output. This can be expressed mathematically as:


(19)
$$\sum_{j=1}^{N}∥O_{j}-t_{j}∥=0$$


There are *β_i_*, *ω_i_*, and *b_i_*_,_ which make:


(20)
$$\sum_{i=1}^{L}\beta_{i}h\left(\omega_{i}\cdot x_{j}+b_{i}\right)=t_{j},\left(j=1\cdots N\right)$$


The matrix can be expressed as:(21)Hβ=T
where the output of the hidden layer node is denoted by:$$H={\left[\begin{array}{ccc} h\left(\omega_{1}\cdot x_{1}+b_{1}\right) & \cdots & h\left(\omega_{L}\cdot x_{L}+b_{L}\right)\\ \vdots & \cdots & \vdots \\ h\left(\omega_{1}\cdot x_{N}+b_{1}\right) & \cdots & h\left(\omega_{L}\cdot x_{N}+b_{L}\right) \end{array}\right]}_{N\times L}$$
and the output weight matrix is represented by *β* and *T* is the target matrix.

Finally, the expression of its solution is as follows: (22)$$\hat{\beta }={\left(H^{T}H\right)}^{-1}H^{T}T$$

### 4.4. SVM (Support Vector Machine)

Support vector machine (SVM) is a supervised machine learning algorithm based on the principle of minimum structural risk [24]. SVM can classify problems into linear and nonlinear types, to solve nonlinear problems, SVM introduces a kernel function *k*(*x,z*), in which radial basis function can map original data into high-dimensional space, and the expression is:(23)$$k(x,z)=\exp\left(-\frac{||x-z|{|}^{2}}{2\sigma^{2}}\right)$$

After introducing the kernel function, the SVM classification decision function is


(24)
$$g\left(x\right)=sign\left(\sum_{i=1}^{n}a_{i}y_{i}k\left(x_{i},x\right)+b\right)$$


To solve the problem whereby some sample points are still inseparable after projection into high-dimensional space, the relaxation variable *ξ_i_* is introduced into the classification model for optimization adjustment, and penalty factor *C* is introduced to the loss part. The new model is obtained by introducing the relaxation variable *ξ_i_* and penalty factor *C*:


(25)
$$\left\{\begin{array}{c} \min_{w,b}\frac{1}{2}{∥w∥}^{2}+C\sum_{i=1}^{n}\xi_{i}\\ y_{i}\left(w^{T}x_{i}+b\right)\geq 1-\xi_{i},\xi_{i}\geq 0,i=1,2,\cdots ,n. \end{array}\right.$$


Convert it to a dual problem:
(26)$$\left\{\begin{array}{c} w_{\max}=\sum_{i=1}^{n}a_{i}-\frac{1}{2}\sum_{i=1,j=1}^{n}a_{i}a_{j}y_{i}y_{j}k\left(x_{i},x_{j}\right)\\ \;s.t.\;\sum_{i=1}^{n}a_{i}y_{i}=0,0\leq a_{i}\leq C,i=1,2,\cdots ,n \end{array}\right.$$
where *a_i_* and *a_j_* represent Lagrange multipliers; *y_i_* and *y_j_* represent the training sample category, where *y_i_* and *y_j_* ∈ {−1, 1}. Penalty factor *C* is used to measure the complexity of the learning machine: if the value of *C* is too large, overfitting is likely to occur, and the SVM model will tend to be complex, with longer running time and reduced operational efficiency; if the value of *C* is too small, the fitting degree of the data sample will be reduced, and the SVM model will be prone to underfitting [25].

## 5. Simulation and Verification of the Platform Model

Firstly, the signal of the gyroscope is obtained, and after obtaining the signal, the output is first decomposed using the VMD algorithm; after that, the signal is decomposed by sample entropy, sample energy, energy entropy, and envelope algorithm for different frequencies bands to obtain the signal features, which constitute the feature dataset; secondly, the training set and test set of feature samples are divided, and the training set is used for training, and the test evaluation is completed using the test set for training.

### 5.1. Simulation

Different types of faults may occur in the gyroscope during long-term operation due to different fault-inducing factors. According to different types, gyroscope faults can be divided into the following types:

Bias fault: this refers to the constant difference between the output signal and the actual value after a certain moment. Its mathematical description form is:(27)$$y_{s}=\left\{\begin{array}{l} y\left(t\right)\;\;\;\;\;\;\;t<t_{s}\\ y\left(t\right)+d\;\;\;t\geq t_{s} \end{array}\right.$$

Block fault: this refers to a moment when the output of the gyroscope is stuck at a constant bias, keeping its output value unchanged. Its mathematical model is:(28)$$y_{s}=\left\{\begin{array}{l} y\left(t\right)\;\;\;\;\;\;\;t<t_{s}\\ y\left(t_{s}\right)\;\;\;t\geq t_{s} \end{array}\right.$$

Drift fault: due to changes in the surrounding environment or internal parameters, such as temperature changes or calibration issues, the output of the gyroscope has an increased constant term. The output value often accumulates additional errors over time. Its mathematical expression is:(29)$$y_{s}=\left\{\begin{array}{l} y\left(t\right)\;\;\;\;\;\;\;\;\;\;\;\;\;t<t_{s}\\ y\left(t\right)+kt\;\;\;\;t\geq t_{s} \end{array}\right.$$
where *k* is the drift rate.

Multiplicative fault: this means that the output signal of the gyro is multiplied with a multiplication factor from a certain point in time, mainly due to scaling errors in the output. Its mathematical expression is:(30)$$y_{s}=\left\{\begin{array}{l} y\left(t\right)\;\;\;\;\;\;\;\;\;t<t_{s}\\ ky\left(t\right)\;\;\;\;\;\;t\geq t_{s} \end{array}\right.$$

Periodic fault: this means that the output signal of the gyroscope is attached to a signal of periodic change from a certain time. Its mathematical expression is:(31)$$y_{s}=\left\{\begin{array}{l} y\left(t\right)\;\;\;\;\;\;\;\;\;\;\;\;\;\;\;\;\;\;\;\;\;\;\;\;\;\;\;\;t<t_{s}\\ y\left(t\right)+square\left(t\right)\;\;\;\;\;\;\;t\geq t_{s} \end{array}\right.$$
where *square*(*t*) is the square wave signal, which is expressed as:(32)$$square\left(t\right)=square\left(t+T_{0}\right)$$
(33)$$square\left(t\right)=\left\{\begin{array}{l} d\;\;\;\;\;\;\;\;0\leq t<\frac{T_{0}}{2}\\ -d\;\;\;\;\frac{T_{0}}{2}<t<T_{0} \end{array}\right.$$

Internal structure fault: the equivalent mass, stiffness, and damping of the gyroscope have changed.

To verify the effectiveness of the proposed dualmass MEMS gyroscope fault diagnosis platform, we conducted simulations to verify it. The Simulink simulation platform was used to generate seven types of signal: normal signals and six fault types of signals, as shown in Figure 6 and Figure 7, where Figure 6 indicates the output form of normal signals and Figure 7 indicates the output form of fault signals. Sixty-seven sets of signals were generated, of which forty sets were used to train the model and twenty-seven sets were used to test the accuracy of the model.

Feature extraction was performed for seven different signals, and classification was performed using six algorithms—ELM [26], SVM [27], KNN [28], NB [29], NN [30], and DTA [31]—and the recognition rate of the classification results is shown in Figure 8, from which we can see that the accuracy of classification recognition using the ELM algorithm was 92.86%, which was able to identify 100% of bias, blocking, drift, period, and internal fault signals, respectively. The accuracy of classification recognition using the SVM algorithm was the same as that of ELM algorithm; the accuracy of classification recognition using the KNN algorithm was 89.29%, whereby it was able to identify 100% of bias, blocking, drift, period and internal fault signals, 50% of normal signals and 75% of multiplicative signals. The accuracy of classification recognition using the NB algorithm was 89.29%, whereby bias, blocking, drift and period signals were identified with 100% probability, and normal, multiplicative and internal fault signals were identified with 75% probability. The accuracy of classification recognition using the NN algorithm was 85.71%, in which bias, blocking, drift and period signals were identified with 100% probability, while 50% of the normal signals were recognized, 75% of the multiplicative signals and 75% of the internal fault signals. Classification recognition using the DTA algorithm was 82.14%, whereby 100% of the bias, blocking, drift and periodic signals, 25% of the normal signals, 75% of the multiplicative signals and 75% of the internal fault signals were recognized. To show the accuracy of identification more visually, histograms were obtained, as shown in Figure 9. From Figure 9, it can be seen that both ELM and SVM have high classification accuracy compared to other fault diagnosis algorithms. The combined analysis shows that the best results were obtained using ELM with the SVM algorithm, and the diagnosis results show the effectiveness and accuracy of the method.

For the trained model, to verify the effectiveness of the training, twenty-eight groups of signals were used for verification in this paper, and the seven types of signals, namely, normal, bias, blocking, drift, multiplicative, periodic, and internal fault, are labeled as the corresponding numbers [1,2,3,4,5,6,7]. Finally, the prediction diagram of the fault diagnosis platform obtained is shown in Figure 10, where the blue dots indicate the actual signal type, and the black lines indicate the recognized signal type. From the figure, it can be seen that only the 16th and 23rd sample point fault types were not accurately identified, while the rest were accurately identified, and the recognition rate of the fault diagnosis platform proposed in this paper is as high as 92.9%, which can effectively and accurately identify the type of signal. Therefore, it can be said that both the training classification accuracy and the test classification accuracy of the fault diagnosis model can reach an equal or higher classification level.

### 5.2. Experiment Verification

To assess the validity and efficacy of the MEMS gyroscope fault diagnosis platform, we conducted a temperature experiment to obtain drift signal data from an actual gyroscope. Figure 11 displays both the gyroscope and the experimental equipment employed in this study. Specifically, we utilized the dual mass-07# gyroscope for our experiment, with detection circuits arranged across three separate PCB boards and connected through the use of metal pins to transmit electronic signals and maintain structural integrity. To minimize electromagnetic interference, the metal case was grounded to the “GND” signal. The first of the three PCB boards served as an interface for processing weak signals, with the structural chip connected to it. The remaining two PCB boards were utilized for the sensing loop and the drive closed loop, respectively [32,33,34].

In our experiment, we utilized a range of equipment to test the feasibility and efficacy of the MEMS gyroscope fault diagnosis platform. This equipment included an Agilent 33220A signal generator, which was capable of generating the test voltage, V_Tes_. Additionally, we employed an Agilent 34401A multimeter and an Agilent DSO7104B oscilloscope to measure and observe signal amplitude and phase. For the power supply, we utilized an Agilent E3631A unit, which provided ±10 V DC voltage and GND. Finally, to create a full range of temperature conditions for measuring the actual bandwidth of the gyroscope, we utilized both a temperature box and a turntable.

The experimental process proceeded as follows: The gyroscope was first turned on and allowed to run for one hour at room temperature. Subsequently, the oven temperature was gradually increased until it reached 60 °C. Then, to ensure a continuous and stable internal temperature in the gyroscope, the temperature should be maintained for one hour for every 10° drop from 60 °C to −40 °C and finally stand for one hour, the test was finished [35,36,37]. The above steps were repeated three times to obtain three groups of experimental results, as shown in Figure 12. For the three groups of drift signals, VMD−SE is used for feature extraction, and the ELM method is used for fault identification. The results of our identification process are presented in Figure 12. It is evident from the figure that the recognition rate based on real data is exceptionally high, reaching 100%. All of the data were successfully recognized, thus providing strong evidence for the reliability and accuracy of the proposed dualmass MEMS gyroscope fault diagnosis platform.

## 6. Conclusions

Due to the inconvenience of gyroscope fabrication costs and fault dataset acquisition, in this study, a new Dualmass MEMS gyroscope fault diagnosis platform was designed. The platform integrates a dualmass MEMS gyroscope structural model and measurement control system, data feature extraction and classification prediction algorithm, and real data feedback verification. The accuracy rate reached up to 92.86% upon simulation training and test set verification, and all the effects of the platform were correctly identified through temperature experiments to obtain real data, indicating that the proposed fault diagnosis platform can accurately and effectively identify fault types.

## Figures and Tables

**Figure 1 micromachines-14-01177-f001:**
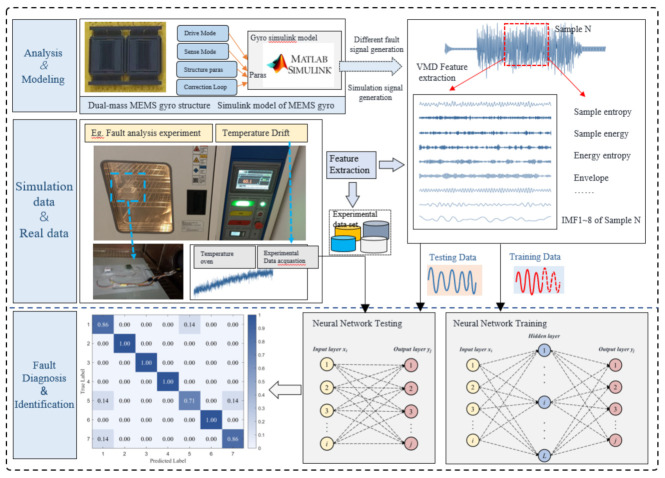
A new dualmass MEMS gyroscope fault diagnosis platform.

**Figure 2 micromachines-14-01177-f002:**
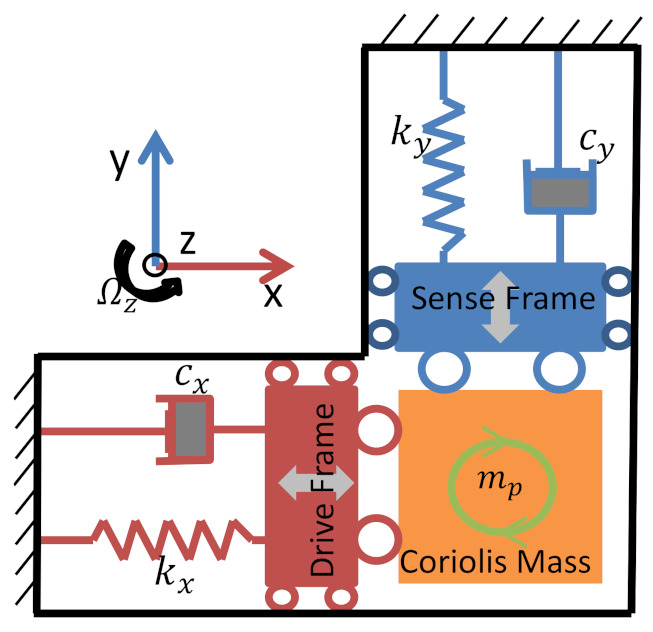
Schematic of fully decoupled structure.

**Figure 3 micromachines-14-01177-f003:**
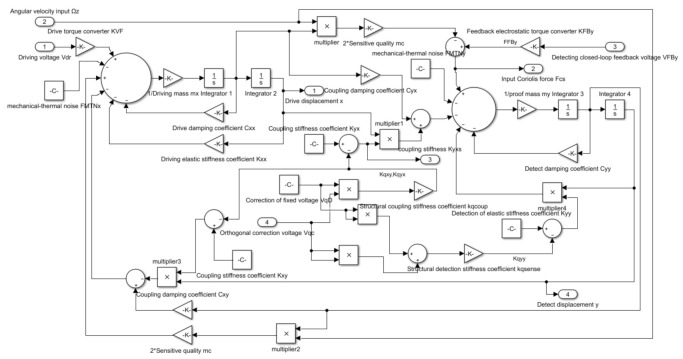
Equivalent model of gyroscope structure.

**Figure 4 micromachines-14-01177-f004:**
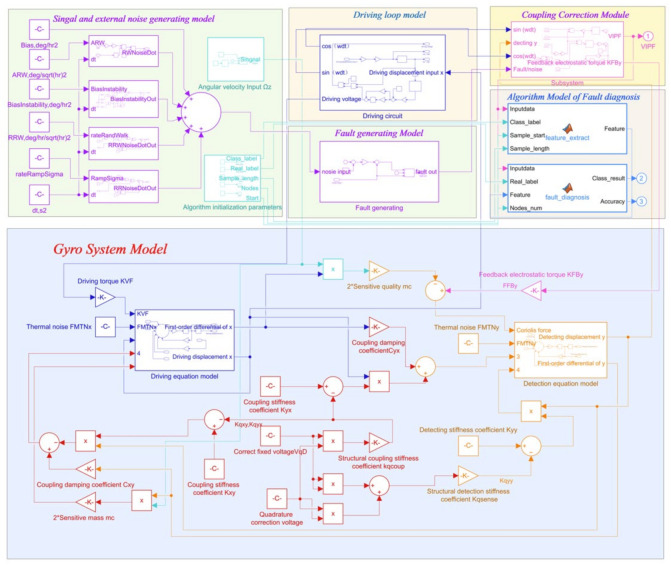
Simulink-based platform model.

**Figure 5 micromachines-14-01177-f005:**
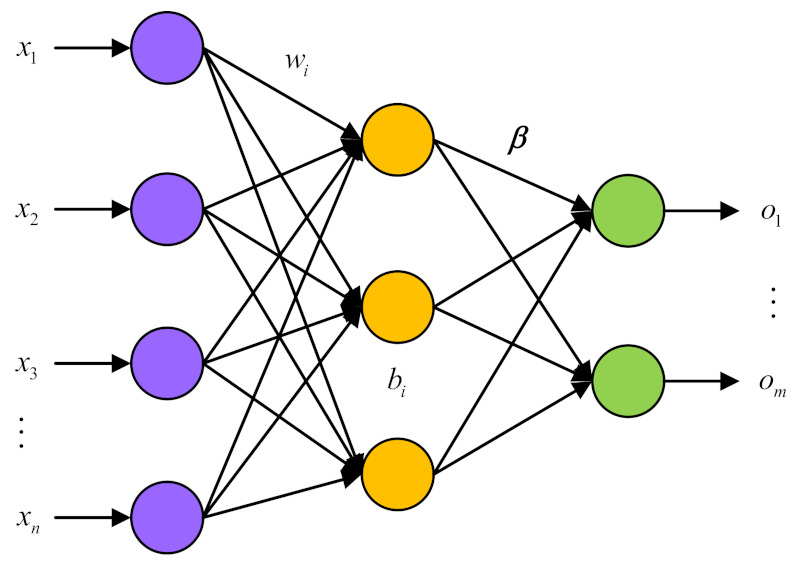
The network model of ELM.

**Figure 6 micromachines-14-01177-f006:**
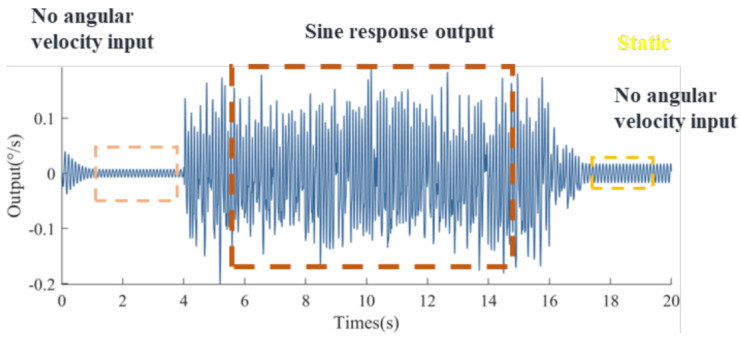
Normal signal of MEMS gyroscope.

**Figure 7 micromachines-14-01177-f007:**
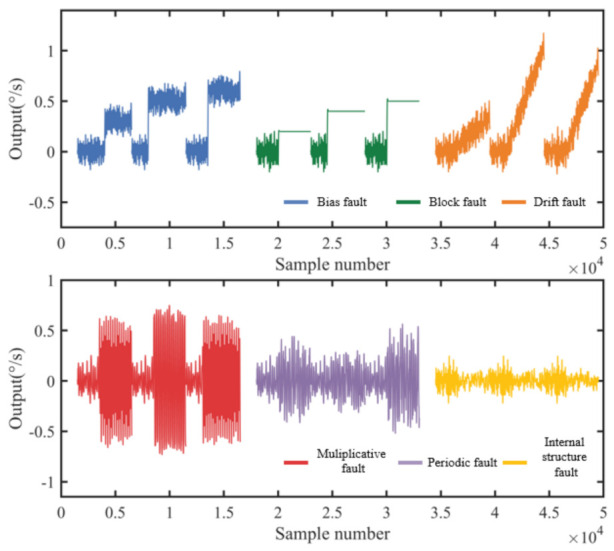
Fault class of the dualmass MEMS gyroscope.

**Figure 8 micromachines-14-01177-f008:**
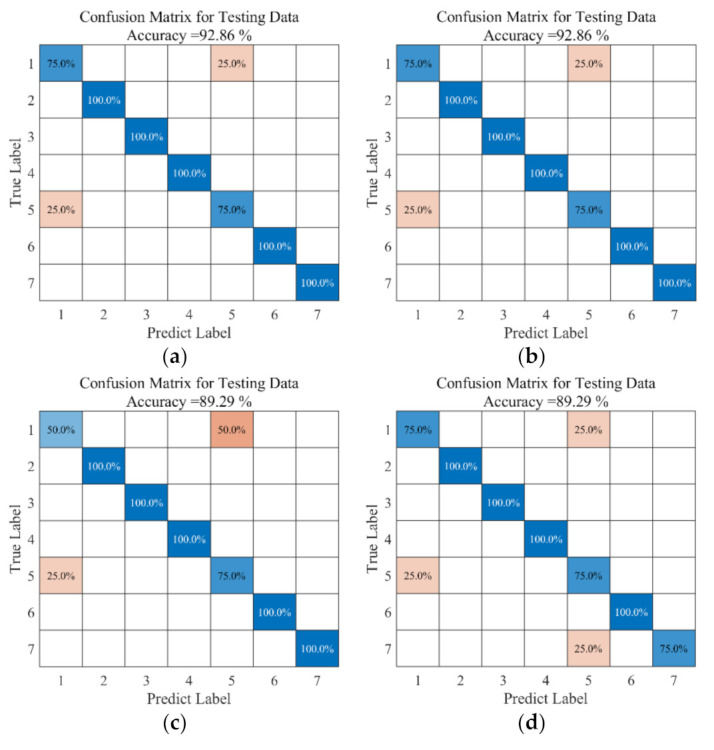
The recognition results of different algorithms: (**a**) ELM; (**b**) SVM; (**c**) KNN; (**d**) NB; (**e**) NN; (**f**) DTA.

**Figure 9 micromachines-14-01177-f009:**
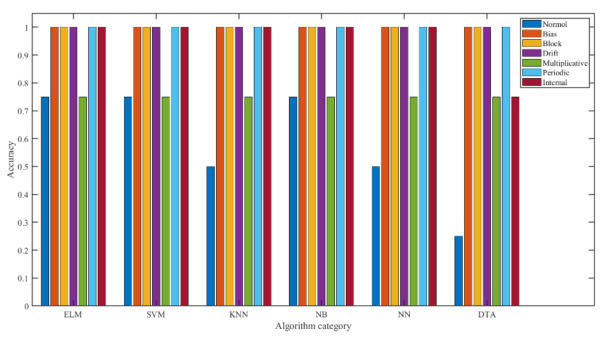
Bar chart of accuracy with seven groups for cross-validation.

**Figure 10 micromachines-14-01177-f010:**
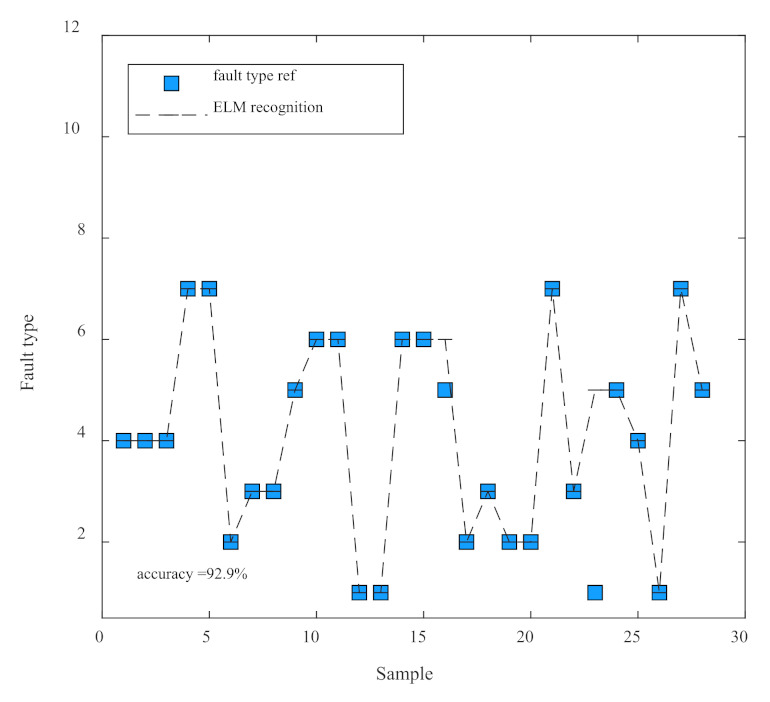
Test results.

**Figure 11 micromachines-14-01177-f011:**
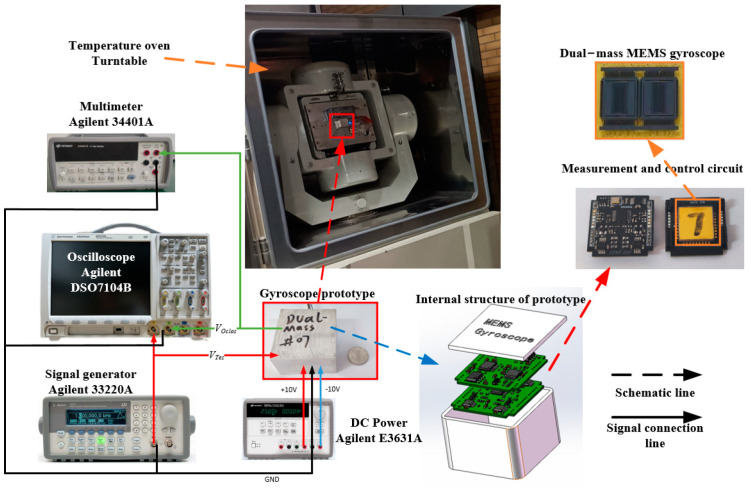
MEMS gyroscope porotype and temperature experiment platform.

**Figure 12 micromachines-14-01177-f012:**
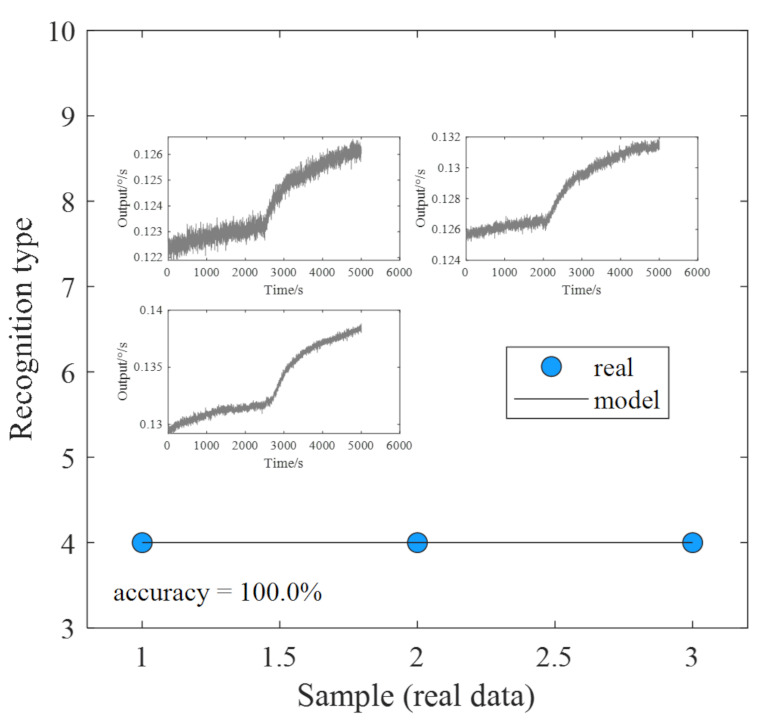
Identification results of real data.

## Data Availability

Not applicable.

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
