# Peer review of "A New Dual-Mass MEMS Gyroscope Fault Diagnosis Platform"

_micromachines, 2023, doi:10.3390/mi14061177_

Round 1

Reviewer 1 Report

In the article titled “A New Dual-Mass MEMS Gyroscope Fault Diagnosis Platform” proposes a dual-mass MEMS gyroscope fault diagnosis platform. However, in order to make this paper more accessible to readers, I think the following changes should be made:

1.     In the introduction, the algorithm is summarized in detail. However, this paper mainly describes the fault diagnosis platform, so it is suggested to increase the overview of the fault diagnosis system, such as adding the description of the fault diagnosis function.

2.     Some figures are too simple to describe, so it is suggested to supplement them appropriately, such as Figure 10. Figure 11 cannot be found in the paper, please note the arrangement of the figures.

3.     There are grammatical problems in this paper, which need the author's attention.

There are grammatical problems in this paper, which need the author's attention.

Reviewer 2 Report

The work proposes a novel MEMS gyroscope fault diagnosis platform for dual mass gyroscopes. The manuscript is well written and supported by all the required details.

1) It will be interesting to add the expected performance of the proposed framework on various other types of MEMS gyroscopes?

The quality of the English language is good, and a minor spell and grammar check is recommended. 
